



# First Reported Detection of a Winter Continental Gamma-Ray Glow in Europe

Jakub Šlegl[1,2], Zbyněk Sokol[3], Petr Pešice[3], Ronald Langer[4], Igor Strhárský[4], Jana Popová[3], Martin Kákona[1,4], Iva Ambrožová[1], and Ondřej Ploc[1]

[1]Nuclear Physics Institute of CAS, Husinec - Řež 130, 250 68 Řež, Czech Republic
[2]Faculty of Nuclear Physics and Physical Engineering of the Czech Technical University in Prague, Břehová 7, 115 19 Prague, Czech Republic
[3]Institute of Atmospheric Physics of CAS, Boční II 1401/1a, 141 00, Prague, Czech Republic
[4]Institute of Experimental Physics of SAV, Watsonova 1935/47, 040 01 Košice, Slovakia

**Correspondence:** Ondřej Ploc (ploc@ujf.cas.cz)

**Abstract.** This study presents the first-ever detection of a winter continental gamma-ray glow in Central Europe, observed during a rare winter thunderstorm on Milešovka hill, Czechia. Unlike typical gamma-ray glow events, which are usually linked to significant electric field increases, this unique observation reveals that no substantial electric field change was recorded during the glow, challenging existing models of thunderstorm-related radiation. The event was captured using a combination of

advanced instruments, including a Ka-band cloud profiler and a high-energy gamma-ray spectrometer, which enabled detailed analysis of the storm's microphysics. The radar data indicated the alignment of ice crystals within the cloud, strongly suggesting the presence of a substantial electric field, despite its weak measurement on the surface. This unexpected decoupling of electric field strength and gamma-ray glow generation opens new avenues for understanding the processes driving high-energy phenomena in thunderstorms. The findings offer valuable insights into winter thunderstorm dynamics in continental climates,

with broader implications for the study of high-energy atmospheric physics.

## 1 Introduction

Since the first observations of radiation coming from thunderclouds made onboard airplanes and balloons (McCarthy and Parks, 1985; Kelley et al., 2015; Kochkin et al., 2017), many other observations were also reported on the ground at locations such as Lomnický štít, Musala, Aragats, Florida, Tibet, and Japan (Kudela et al., 2017; Chum et al., 2020; Šlegl et al., 2022;

Chilingarian et al., 2021c, b, 2016; Chilingarian and Mkrtchyan, 2012; Chilingarian et al., 2010; Williams et al., 2022; Tsuchiya et al., 2012, 2007, 2009, 2011; Torii et al., 2009, 2011; Wada et al., 2021b, a, 2019, 2018, 2023).

Gamma-ray glows, also called long bursts (Tsuchiya et al., 2009; Torii et al., 2011) or thunderstorm ground enhancements (Chilingarian et al., 2011), are minute-scale long enhancements in the radiation environment with passing thunderclouds. The mechanisms of their origin are attributed to the process called relativistic runaway electron avalanche (RREA) (Gurevich et al.,

1992) and to the modification of the energy spectra (MOS) (Chilingarian and Mkrtchyan, 2012) created between regions of opposite charge strong enough, with at least one situated in the cloud.



This region of strong electric field (charged region) moves with the cloud and creates typical bell-shaped enhancements in the count rates observed by ground detectors (Wada et al., 2021b). The peak is detected at the moment of the shortest distance between the detector and the center of the active (charged) region.

In some cases, such as the presented event, the gamma-ray glow is terminated by discharge. This discharge reduces at least one of the charged regions and makes the RREA stop. Many such cases have been reported (Tsuchiya et al., 2013; Chilingarian et al., 2015, 2017), with discharges sometimes starting far from the detected gamma-ray glow where the intra-cloud (IC) leader passed by (Wada et al., 2018, 2019).

Winter thunderstorms are much rarer, especially in continental Europe. During 4 years of continuous measurements on top 30   of the Milešovka hill, only a few summer season gamma-ray glows were observed (Kolmašová et al., 2022), and none during winter until the event observed on February 4, 2022, which is presented here.

For the study of thunderstorms, the observatory on the Milešovka hill (837 m a.s.l.) was equipped with a unique set of measuring instruments for the detection of ionizing radiation, meteorological and climatological observations, including the cloud profiler (Ka-band).

## 35   2   Detectors and Data

Milešovka station (50.55° N, 13.93° E), Czech Republic, is a professional meteorological station with a 24/7 service located at the top of the Milešovka hill at an altitude of 836 m a.s.l. It exceeds the surrounding area by about 300 meters.

Apart from standard meteorological and climatological measuring instruments (thermometer, barometer, rain gauges, hygrometer, wind speed and direction meters), the station is also equipped with a ceilometer, disdrometer, Ka-band cloud profiler, 40   and radiation detectors SEVAN with a gamma-ray spectrometer Georadis RT-56.

The Georadis **RT-56** spectrometer includes a 3" × 3" cylinder BGO crystal coupled with a photomultiplier tube (PMT) and electronics for data acquisition. The primary spectrum starts at 0 and spans up to 4.5 MeV with 1500 channels separated by 3 keV. The calibration is regular and automatic, based on naturally abundant isotopes.

The cosmic mode of the spectrometer stores information on deposited energies higher than the primary spectrum, while the 45   measuring mode is the Time-over-Threshold mode, with a threshold of 4.05 MeV calibrated by peak shape fits. The RT-56 spectrometer also calculates the number of pileups per second and contains a GPS module. The spectrometer is able to provide a timestamp of up to 4000 particles every second in a selected energy interval. In the case of the winter thunderstorm on February 4, 2022, studied in this paper, it was set up to 4.5 MeV.

Another important feature of the detector are its internal batteries which ensure uninterrupted measurements during thun-50   derstorms when blackouts are common, especially in mountainous areas. The batteries of the detector at the Milešovka station can maintain the full detector operation for up to one day.

The second ionizing radiation detector used in this study is the **SEVAN** detector system (Chilingarian et al., 2009), which consists of three plastic scintillator detectors, each coupled with one PMT. The upper one as well as the bottom one are set up from four plastic scintillator slabs creating a square with one-meter long sides. The middle one, separated from the top one and



the bottom one by a 5 cm thick lead, consists of five stacked slabs. The energy threshold of the upper channel of SEVAN is expected to be approximately 7 MeV.

This set-up allows, together with fast electronics, to estimate particle types as well as their intensity. For example, coincidence 100 (response in the upper channel while no response in the bottom and middle channels in a narrow time interval) shows interactions of most probably photon or electron. Coincidences 101 and 111 show most probably muons as they pass

through both lead layers.

**Electric field mill Boltek EFM-100C** manufactured by the Boltec company measures the electrostatic field in a vertical direction. It is oriented downwards (inverted position) to minimize the precipitation noise. Its negative values mean that the electric field's direction is upwards and electrons are accelerated downwards. The sampling frequency is 20 Hz.

**Cloud profiler MIRA 35c** at the Milešovka station is a vertically oriented Doppler polarimetric radar manufactured by

the METEK GmbH company. It operates within Ka-band with a center frequency of $35.12 \pm 0.1$ GHz and a peak power of 2.5 kW. The temporal resolution is 2 s and vertical resolution includes 509 gates separated by 28.8 m. More technical details are given in Sokol et al. (2020) and Kolmašová et al. (2022). Data provided by the cloud profiler are radar reflectivity (Ze), Doppler vertical velocity, spectrum width, Linear Depolarization Ratio (LDR), and co-polar correlation coefficient (RHO). Postprocessing of the data was carried out with the methods described in Sokol et al. (2018) which also calculates vertical air

velocity (Va) and provides hydrometeor classification.

**Disdrometer Thies Laser Precipitation Monitor** with an infrared laser beam is capable of distinguishing the type of hydrometeors by their size and fall speed.

Lightning data recorded by the **EUCLID** (European Cooperation for Lightning Detection, https://www.euclid.org/) network by BLIDS (Blitz Informationsdienst von Siemens), used in this paper, contain records of the time (with the accuracy of ms),

position, type (Cloud-to-Cloud, CC; or Cloud-to-Ground, CG), estimated peak current (in kA including its polarity), and quality (good/bad) of registered discharges. It should be mentioned that all lightning data for the studied winter storm were of good quality.

**Blitzortung**, a lightning detection network for localization of atmospheric discharges with very low-frequency receivers (3 to 30 kHz), uses the Time of Arrival and Time of Group Arrival techniques to register lightning discharges (Wanke, 2010). As

the Milešovka hill is in the area of a dense coverage of antenna stations, we believe that the accuracy of this network is within 1 km in the area, as the Blitzortung project claims.

## 2.1 Meteorological Situation

Winter thunderstorms in the Czech Republic, a landlocked country, are rare. Munzar and Franc (2003) mentioned a rise in the number of winter thunderstorms over the whole Czech Republic but not more than a couple per year. Unlike summer storms,

the winter thunderstorms cause less economic damage but are more unpredictable compared to summer thunderstorms.

Munzar and Franc (2003) also related the occurrence of winter thunderstorms over the Czech Republic with cold fronts. This was also the case on February 4, 2022 (see Figure 1), when gamma-ray glow was measured by two detectors. The cold front



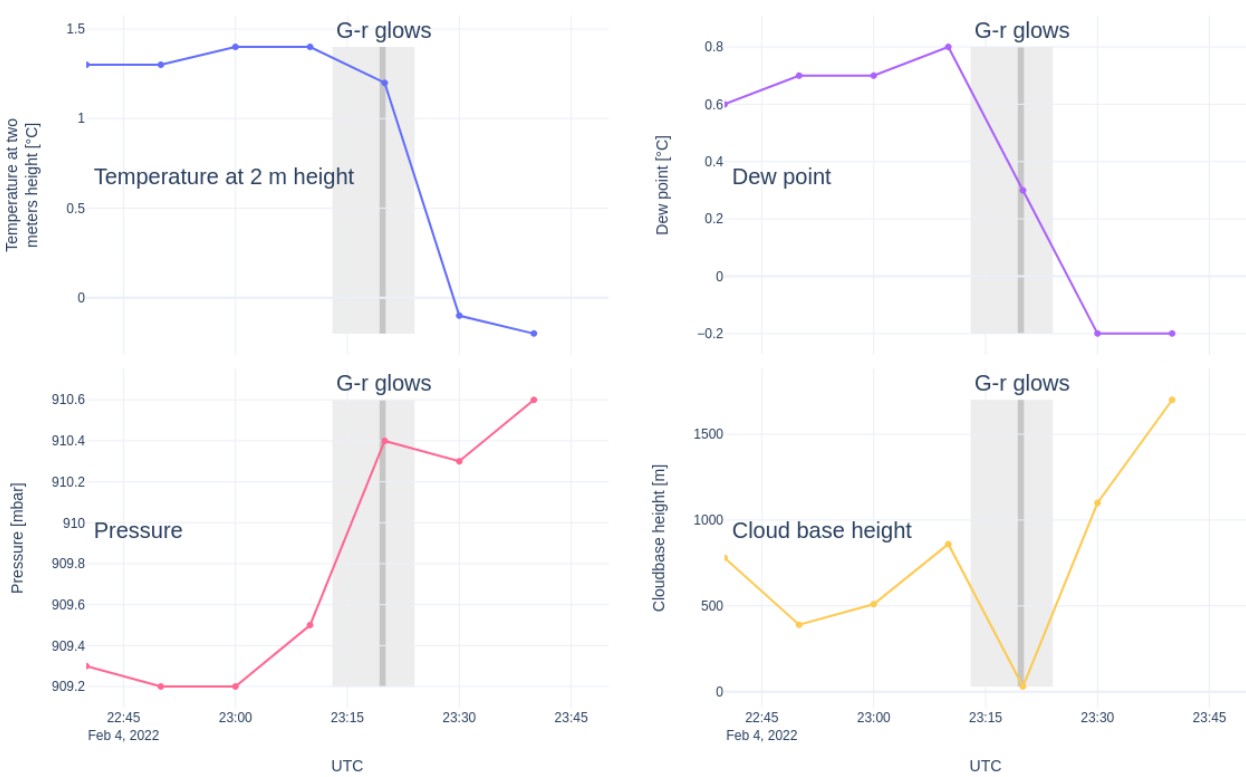

**Figure 1.** Change in 2 m temperature (top left), dew point at 2 m (top right), pressure (bottom left), and cloud base height (bottom right) during the thunderstorm on February 4, 2022. Note that the cloud base height is at the lowest measurable value at 23:20 UTC. The actual cloud base was below the observation site. Solid lines only connect points of measurements for better readability. Gamma-ray glows are represented by gray areas.

with temperatures below zero passes the Milešovka hill with a cloud base below the station at the moment of the gamma-ray glow event.

According to the Blitzortung network data, the thunderstorm lightning activity started in Germany, near Dortmund, around 17:30 UTC. It crossed Germany and reached the German-Czech borders and the Ore Mountains at 22:50 UTC, then crossed the Milešovka station and eventually dissipated after the last discharge east of the Milešovka hill at 23:23 UTC (see Figure 2).

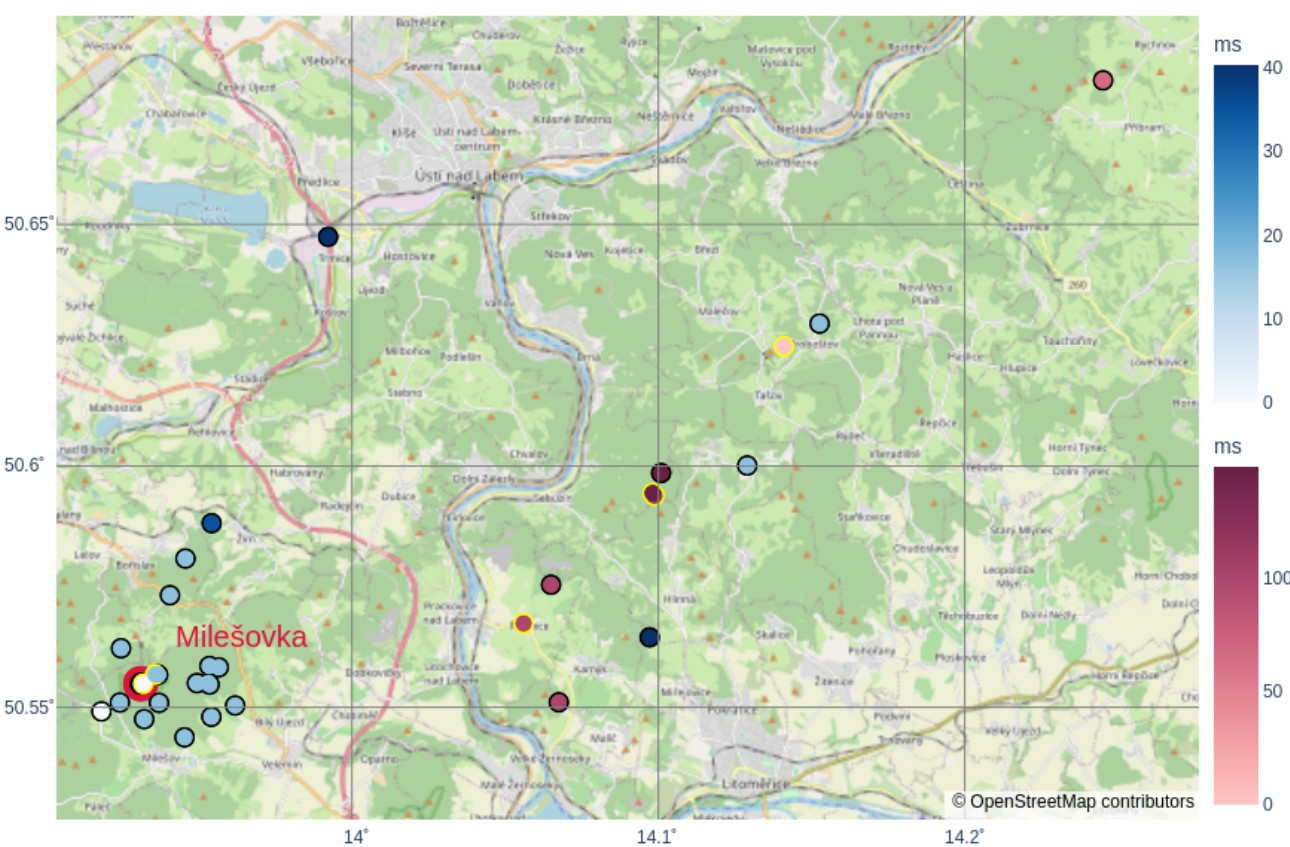

**Figure 2.** Discharges detected by Blitzortung (black-bordered circles) and EUCLID (yellow-bordered circles) at the time of the end of the gamma-ray glow (23:20:08.147-187, blue scale) and 3 minutes later (23:23:03.300-450, reddish scale). Color scales show time in milliseconds from the first discharge. Both networks indicate the beginning of the gamma-ray ending discharge very near to the Milešovka station depicted by a red-bordered circle. © OpenStreetMap contributors 2024. Distributed under the Open Data Commons Open Database License (ODbL) v1.0.





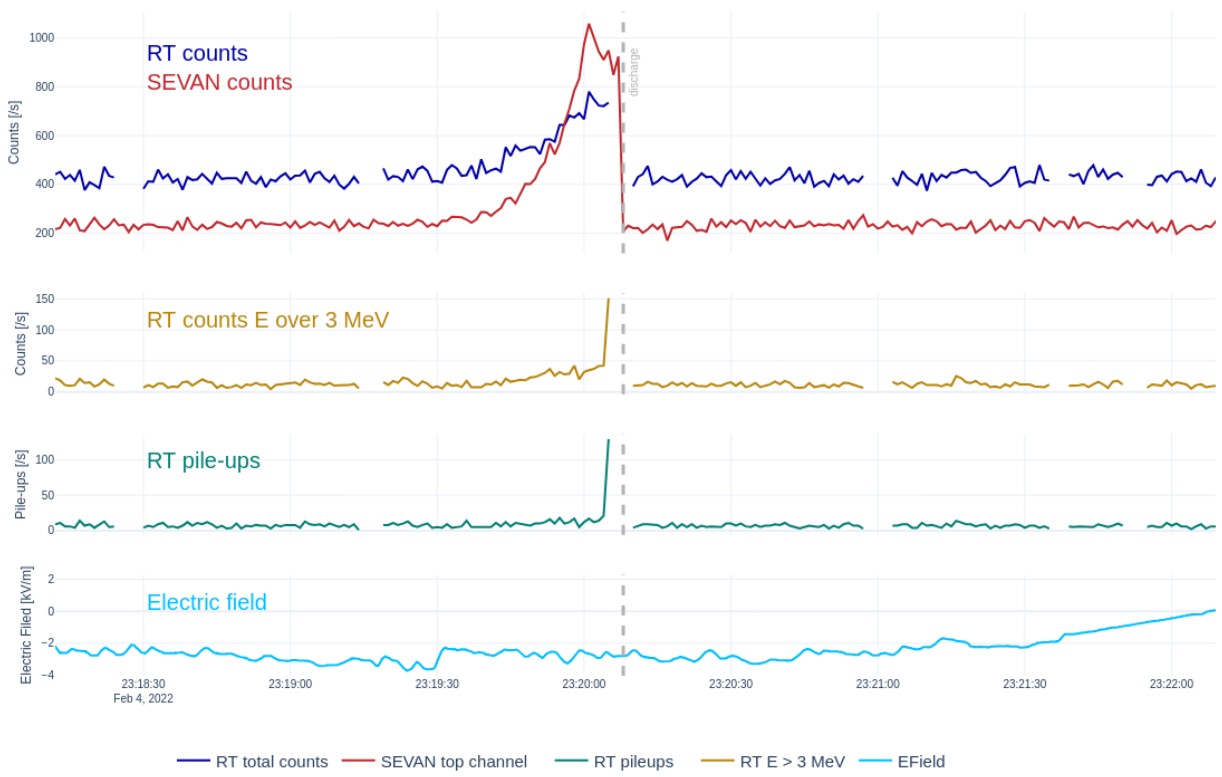

**Figure 3.** Gamma-ray glow observed by both SEVAN and RT-56 spectrometer. SEVAN top channel (red), RT-56 all counts/s (blue), RT-56 counts/s of photons with energy above 3 MeV (yellow), RT-56 detected pile-ups (green), near surface electric field measured by Boltek EFM-100c (cyan). Note that the gaps in data of RT-56 were caused by a software error in timing from GPS.

## 3 Results

### 3.1 Discharge Ending Gamma-Ray Glow

During the gamma-ray glow, there was only a weak electric field measured by the electric field mill. The electric field data also surprisingly show that there is no sudden detected change of electric field with the discharge. Usually during the discharge, the data from electric field mills show a sudden change in the electric field towards the opposite polarity with rapid exponential restitution, but that was not the case during the winter thunderstorm on February 4, 2022. This could be due to some frost on the electric field mill and its location close to vegetation (trees are in close vicinity of the electric field mill).

The discharge was accompanied by strong light and sound emissions. The observer on duty at the Milešovka station reported no observable time difference between the lightning and the thunder. This shows the extreme proximity of the discharge to the



station, as supported by Blitzortung and EUCLID data. Shortly after the discharge, the power generator automatically started, suggesting a direct hit to the facility or a nearby strong electric impulse that caused a power outage.

The EUCLID network reported two discharges (See Figure 2) very close to the station. Blitzortung recorded multiple dis-
charges within a radius of 11 km from the station but noted that they began near the station as well. The discharge, ending the gamma-ray glow, lasted for about 40 ms (23:20:08.147-187 UTC). According to the lightning monitoring networks (both EUCLID and Blitzortung), the discharge occurred in very close proximity to the Milešovka station. EUCLID even showed a horizontal distance of 50 m from the station, while it was 1.1 km in the case of Blitzortung. Both networks recorded the same time of the discharge and indicated the continuation of electromagnetic activity at 23:20:08.163. EUCLID showed one more
discharge, very close to the Milešovka station; however, Blitzortung recorded 21 discharges in the vicinity of the station (within a 3 km distance) and two further, 15 and 18 km away from the station, respectively. The following discharges were observed by Blitzortung only (one at 23:20:08.182 and two at 23:20:08.187).

For a detailed analysis of the thunderstorm on February 4, 2022, please refer to Popová et al. (2023).

### 3.2 Electric Field

The weak electric field measured during the gamma-ray glow (see Figure 3) can be explained in several ways. One possibility is that the charge region was partially neutralized by an opposite charge, resulting in an integrated value measured beneath the cloud that shows only low values. Such an arrangement could occur, for example, if the charge is oriented vertically rather than horizontally, as indicated by calculations in Popová et al. (2023). Alternatively, this finding could suggest a more complex charge structure, which aligns with previous studies Brothers et al. (2018). Such a complex structure may lead to ground
measurements failing to capture the strong fields present in the upper cloud layers, thereby explaining the weak electric field recorded during the gamma-ray glow.

### 3.3 Cloud Profiler Data and Disdrometer

The thunderstorm passing the Milešovka station on February 4, 2022, was unique in its height. The cloud base was unusually low—at the altitude of the hilltop or even lower—and the cloud top was also low. As visible in the data from the cloud profiler
(Figure 4), the height of the cloud top did not exceed 3500 m above the hilltop. The cloud top height was also estimated using satellite measurements of Meteosat Second Generation (https://www.eumetsat.int/meteosat-second-generation), not shown here. LDR and RHO values (Figure 4) show that the height of the thundercloud reflecting back the signal was even smaller, reaching a height of only 2500 m and only 1500 m.

Throughout the storm, we observed a low-intensity gamma-ray glow that started at the moment of a higher reflectivity region
with Ze above 15 dBZ and ended when there were very low values of Ze (lower than -15 dBZ; Figure 4 - Ze). The end of the low-intensity gamma-ray glow roughly correlates with the last pixel of graupel and hail (GH), see Figure 5, as deduced from the fitted Gaussian curve, but since the excess of the low-intensity gamma-ray glow was low above the background, its end is not very recognizable and is smeared in the fluctuation of the background. During the low-intensity gamma-ray glow, the cloud contained graupel and hail with a sparse mix of cloud water and raindrops (Figure 5). The classification algorithm has



one class for both hail and graupel. Due to the lower values of the measured Ze, it can be concluded that hail probably did not occur.

    During the storm, we also observed a shorter strong gamma-ray glow, which ended by the discharge. During the strong gamma-ray glow, Ze was above 20 dBZ in the lowest 1500 m of the cloud, and the cloud contained graupel and hail with a sparse mix of cloud water and raindrops, as in the case of the low-intensity gamma-ray glow. These findings are consistent

with ranges previously calculated by Chilingarian et al. (2021b); Diniz et al. (2022).

    The polarimetric feature of the cloud profiler can even reveal areas of aligned ice crystals that change the reflection of the radar signal. This alignment is probably caused by the ambient electric field, which can be used to identify strong electric fields in clouds (Melnikov et al., 2019). Such areas are visible in Figure 4 - LDR, where LDR values higher than -20 dBZ indicate the alignment of ice crystals in the area above the melting layer, which shows the highest LDR values at about 250

m above the radar. These areas are even more pronounced in the projection of RHO; the higher the number, the higher the specific reflectivity leading to a conclusion on intensified electric fields (Melnikov et al., 2019). The most interesting area is just after the discharge (discharge depicted by the dashed line in Figure 4). The narrow area of higher values of RHO right after the discharge spanned from the top to almost the bottom of the cloud. This could confirm the previous existence of an area of intensified electric field that aligned the ice crystals and also caused the gamma-ray glow observed on the ground.

Around the gamma-ray glow event, the disdrometer did not detect any hail. On the other hand, graupel and snow grains were recorded right before the ionizing radiation enhancement. The existence of falling snow grains was also recorded by the observer on duty and agrees well with the observations in other studies (Chilingarian et al., 2021a; Wada et al., 2021b).

### 3.4   Strong Gamma-Ray Glow

The strong gamma-ray glow was detected with both the SEVAN and RT-56 spectrometer. This indicates high reliability of the

results, as both detectors employ different electronics. The SEVAN detector shows the rise in the upper channel only, which indicates a low presence of high-energy photons that are able to penetrate the 5 cm thick layer of lead and thus would have been visible in the two other channels (middle and bottom one).

    The rise in the upper SEVAN channel lasted for 30 seconds and reached a maximum of 835 counts per second (i.e., 375% of the normal level). The peak occurred 8 seconds before the end of the discharge, and the gamma-ray glow ended due to the

lightning discharge between 20:20:08.147 and 20:20:08.187, according to the Blitzortung network.

    The rise on RT-56 was 346 counts per second (i.e., 83% of the normal level). The spectrometer experienced issues with time determination, but we were able to recover most of the gamma-ray glow data. The data acquisition stopped 3 seconds before the discharge, and at that time, a huge number of pileups was detected (125 compared to approximately 7 when there was no gamma-ray glow). The exact pileup peak and its time are uncertain as it might have been reached during the data acquisition

gap. High numbers of pileups per second were also observed by this detector during other events when a lightning discharge was recorded in close proximity to the detector, e.g., 2022-06-24, 2023-07-29, and 2024-07-10. Never in such a high number, though. Whether the pile-up peak can be a detection of stepped leader x-ray pulses, terrestrial gamma-ray flash, or an electronic artifact remains a question.





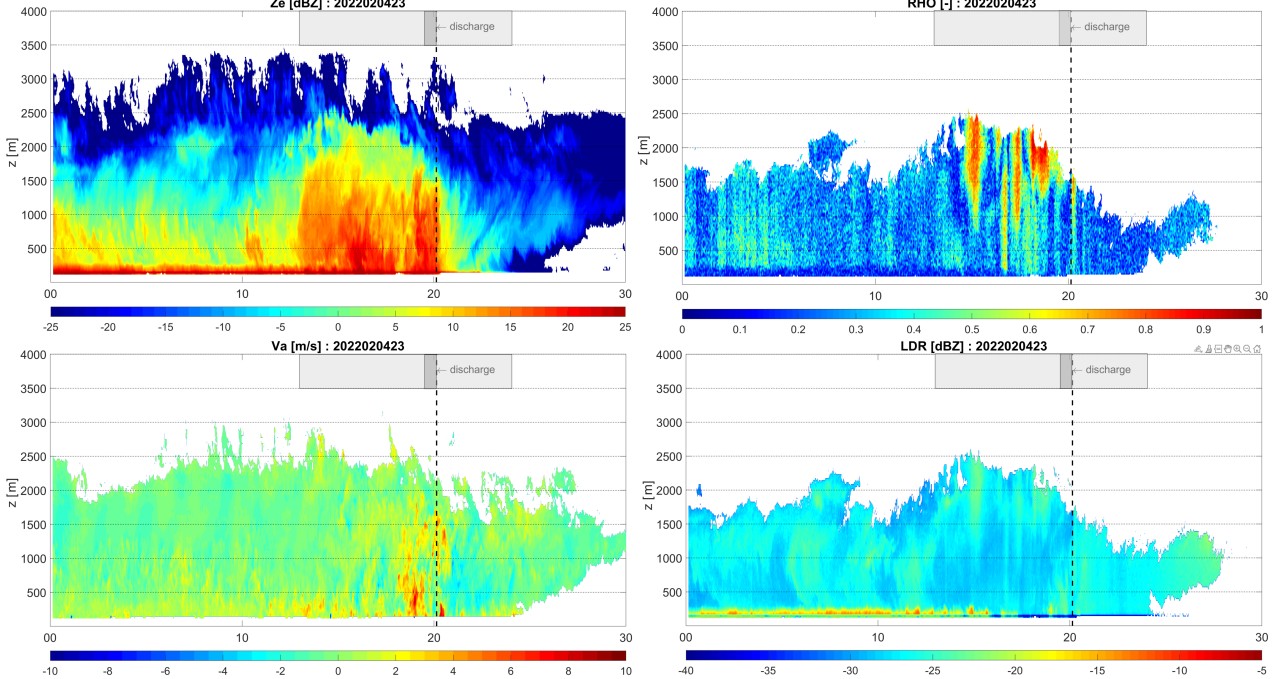

**Figure 4.** *Ze*, *LDR*, *Va*, and *RHO* measured by the vertical profiler MIRA 35c from 23:00 to 23:30 UTC (horizontal axis). The vertical dashed line shows the time of the recorded discharge. Gray areas show both gamma-ray glows: low-intensity gamma-ray glow (light gray), high-intensity gamma-ray glow ended by the discharge (dark gray). For more details see Popová et al. (2023).

As visible in Figure 7, the count rates follow the Gaussian curve up to the maximum of the gamma-ray glow, being still
at the beginning of the Gaussian curve. However, we believe that our fitting is relevant because the fitted curve is similar to those previously observed in e.g., (Šlegl et al., 2022; Wada et al., 2019). If the gamma-ray glow had not been interrupted by the discharge and had continued following the Gaussian curve, the duration of the peak would be roughly 2.5 minutes, which aligns with the previous observations and the peak count rates of 5552 counts/s for SEVAN and 1448 counts/s for RT-56 with subtracted background, considering a wind speed of 10 m/s and a cloud base height less than 30 meters. The RT-56 fit differs
from the SEVAN fit in the time of the maximum, but only by 1 second.

The continuation of the gamma-ray glow, if not terminated by the discharge, would correlate to the echo of aligned ice crystals by the electric field, as shown by the data from the Cloud profiler (Figure 4 - RHO). The continuation of such alignment of ice crystals in the upper part of the cloud can be explained by residually charged areas in the upper part of the cloud, above which the ice crystals were still caught in the electric field. Either the stroke did not discharge the upper charged region totally
or not at all. This suggests the possibility of a horizontal discharge that left the upper charged region intact.



**Figure 5.** Hydrometeor classification using cloud profiler data based on Sokol et al. (2018). CR - Cloud and Raindrops, IS - Ice and Snow, GH - Graupel and Hail. Note sparse small gray areas (CR+GH) within the blue area (GH). For more details see Popová et al. (2023).





**Figure 6.** Disdrometer data around gamma-ray glows (gray vertical areas - light gray for the low-intensity glow, dark gray for the strong glow) shown as number of particles per minute: Number of all particles, Particles < 0.15 m/s, Solid precipitation, Big graupel, Small graupel, Snow grains, Rain, Small rain, Drizzle, no hydrometeor. The horizontal axis is the time in UTC.



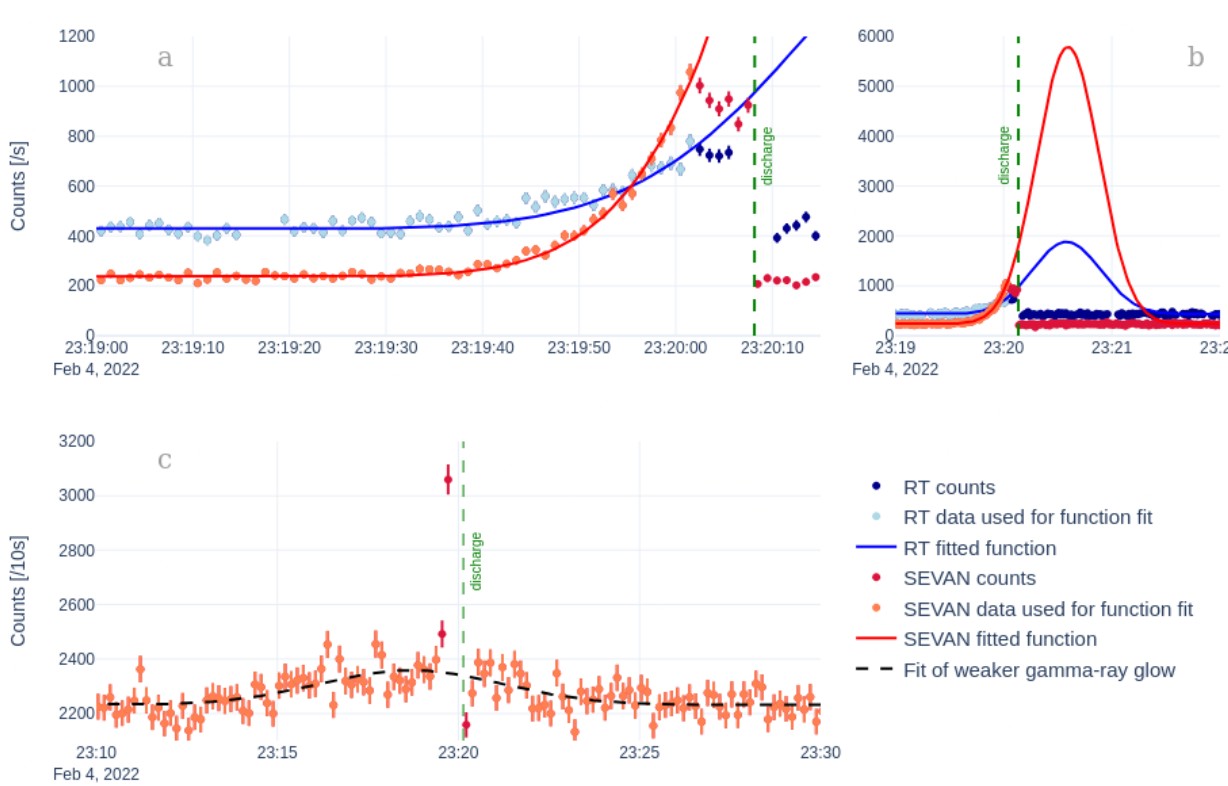

**Figure 7.** Counts measured by RT-56 (blue) and by the upper channel of SEVAN (red). Panel (a) shows a big gamma-ray glow fitted with the best fit function (light markers depict data used for the function fitting, while dark markers represent data not used for the fitting). For RT-56, the data were fitted with a simple Gauss function, whereas for SEVAN, we combined two Gauss functions. Panel (b) represents the expected peak without discharge as it results from the best fit functions. Panel (c) shows SEVAN data during the low-intensity gamma-ray glow and its best fit function in 10 s summed bins (light markers represent data used for the function fitting, while dark markers display data not used for function fitting). Note that two markers of the strong gamma-ray glow are out of the y-axis range of the graph.





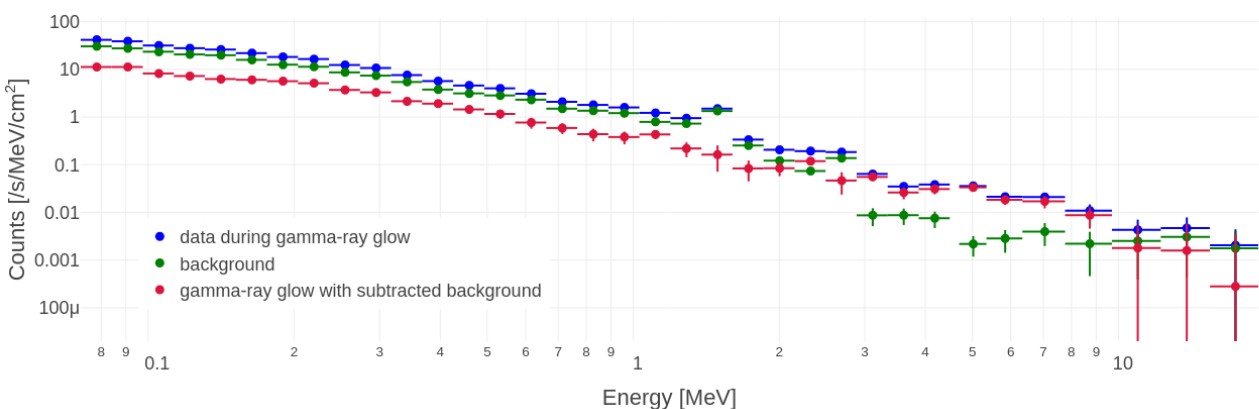

**Figure 8.** Gamma-ray spectrum measured by the RT-56 spectrometer: data during gamma-ray glow - blue (23:19:38 - 23:20:08 UTC), background - green (23:19:00 - 23:19:30 UTC), gamma-ray glow with subtracted background - red. Vertical error bars represent 1 $\sigma$ error.

A question arises about the last 7 seconds (23:20:01.0-23:20:08.0) that preceded the discharge. As it does not follow the Gaussian curve, some mechanism must have prevented the gamma-ray glow from its full development. Such a sharp peak with a decrease before the end of the discharge was also observed by Wada et al. (2019).

### 3.4.1 Gamma-Ray Glow Spectrum

The gamma-ray glow spectrum detected by the spectrometer RT-56 follows the expected curve. Since the statistics at the very high energies (10-20 MeV) are poor, we can state that we detected an excess of photons with energies up to 10 MeV (Figure 8).

## 4 Conclusions

The observation of a winter continental gamma-ray glow on Milešovka hill, Czechia, provides novel insights into thunderstorm-related radiation phenomena in a rarely studied environment. The findings reveal several important points:

1. **First Described Winter Gamma-Ray Glow in Continental Europe:** This event marks the first documented observation of a winter gamma-ray glow in Central Europe, a region where such phenomena are uncommon due to the rarity of winter thunderstorms with suitable conditions.

   2. **Unexpected Weak Electric Field During the Glow:** Despite the clear detection of high-energy gamma radiation, the corresponding electric field measured at the surface remained weak, challenging existing models that associate gamma-195   ray glows directly with significant electric field enhancements.

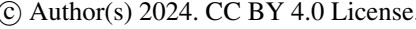

3. **Role of Ice Crystal Alignment in Radiation Generation:** Radar observations showed clear alignment of ice crystals in the cloud, indicative of a strong electric field higher up, even though this was not reflected in surface measurements. This suggests that strong localized electric fields within the cloud, rather than at the ground level, play a crucial role in generating gamma-ray glows.

4. **Unique Meteorological Conditions Contributing to the Event:** The exceptionally low cloud base and the relatively shallow vertical extent of the cloud allowed gamma radiation to reach the ground. This highlights how specific meteorological settings—like a low cloud base and limited cloud height—can influence the occurrence and detection of gamma-ray glows.

These results challenge conventional understanding and suggest the need for revised models that account for decoupling be-
tween ground-level electric fields and high-energy radiation processes within clouds. The findings also underscore the value of combining multiple observational tools, such as cloud radar and an ionizing radiation spectrometer, to unravel the complexities of thunderstorm dynamics and radiation generation in different climatic conditions.

*Data availability.* The cloud profiler data are described in Popova2023, The disdrometer, meteorological, SEVAN, and RT-56 data are available at TBA before publication. link for reviewers https://data.mendeley.com/preview/c3tn4877gj?a=512b87b6-468b-4b99-8b83-754b96986d6f
Graphs of latter mentioned data where produced by plotly Inc. (2015)

*Author contributions.* **J. Šlegl:** conceptualization, investigation, software, data curation, resources, visualisation, writing - original draft **Z. Sokol:** funding acquisition, software, methodology, visualization, writing - review & editing **P. Pešice:** investigation **R. Langer:** investigation **I. Strhárský:** software, resources **J. Popová:** resources, writing - review & editing **M. Kákona:** funding acquisition, resources, validation, writing - review & editing **I. Ambrožová:** supervision, writing - review & editing **O. Ploc:** supervision, funding acquisition, writing - review
& editing

*Competing interests.* The authors declare that they have no known competing financial interests or personal relationships that could have appeared to influence the work reported in this paper.

*Acknowledgements.* This research was supported by the Johannes Amos Comenius Programme (OP JAC), project No. CZ.02.01.01/00/22_008/0004605 "Natural and anthropogenic georisks".



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
