# Peer review of "First Reported Detection of a Winter Continental Gamma-Ray Glow in Europe"

_EGUsphere, 2024_

## Author Response (AR1)

RC1

General (major) comments

This is an intriguing paper that claims to have captured the first gamma-ray glow in Europe during the winter season.

The paper discusses the relationship between the gamma-ray glow and other observations, such as rainfall, electric fields, and radio waves. I believe this work will be of significant interest to specialists in the relevant field.

However, there are some unclear points that may need clarification or correction. Please consider my comments below.

*Thank you for your in-depth reading. We included your suggestions to the text.*

[1] It would help the reader's understanding if the authors could explain why gamma-ray glows have not been observed in the Europe continent during the winter until now. What made it possible to observe one this time?

Was it due to specific meteorological conditions, or had there simply been no prior observations conducted in winter?

*The title claim first reported a winter thunderstorm with a detected gamma-ray glow in Europe. There were gamma-ray glows/TGEs reported in Aragats, Armenia but there can be a dispute about whether Armenia is part of Europe or not (geophysically vs. geopolitically).*

*Winter gamma-ray glows were also detected on Lomnicky stit, but were not reported. A comprehensive study is in preparation. Therefore, we claimed the first "reported" winter gamma-ray glow.*

*Following text was added to the introduction about measurements on Milešovka hill:*

*This is probably caused by several reasons: winter continental thunderstorms are rare (there are known occurring winter thunderstorms near the coast of Japan, near the Great Lakes and in the Mediterranean, all connected to water), they are of low intensity (low flash rate), and their cloud base is higher than in Japan, so that the photons from bremsstrahlung is attenuated bellow detectable level. Other reason is that the measurements of ambient ionizing radiation were not suitable for such observations with too long acquisition intervals (10 minutes for Czech radiation monitoring system - Slegl et al. 2020).*

[2] Does the situation where the cloud base is below the observation station imply that the station is within the cloud?

Additionally, since the cloud base can be determined through a simple calculation if the temperature and dew point are known,

I wonder if this approach has been attempted. Furthermore, what is the measurement error of the ceilometer?

Even when accounting for this error, can it be conclusively stated that the cloud base

was below the observatory?

Based on these considerations, please justify the claim that the glow was detected when the clouds were below.

*The error of the ceilometer measurement is 25 meters, or 5% over 2000 meters. The lowest value the ceilometer gives is 30 meters, which usually means that the instrument is in a cloud with a base below the station. This can be verified by visibility measurements of PWD (Present Weather Detector) or by the SYNOP message sent by observers and giving fog at the station in that case.*

[3] L99 "This could be due to some frost on the electric field mill and its location close to vegetation (trees are in close vicinity of the electric field mill)."

Is this true for this observation?

The authors also mention that there are several lightning discharges occurring before the glow around L104-L112, but there are no electric field variations related to them at all, right? If so, doesn't those mean that the electric-field mill data is unreliable for this entire observation? Please explain this thoroughly, as it pertains to the basis of the paper that there were no electric field fluctuations in this area, but gamma ray glow was observed.

*Yes, after consideration the data from the electric field mill were removed from the text. We experienced in some other cases that nearby flash was not detected on the electric field mill (different spot and different electric field mill than on Milešovka), but after further discussions, we decided to remove the data completely.*

Specific (minor) comments

[1]Could the author please include the difference between Czech time and Universal Standard Time (UTC) somewhere, so that the local time can be clearly understood?

*A sentence was added after the first mention of time: Throughout the paper, we will use time in UTC, local time (CET) is UTC+01.*

[2] I am uncertain which of the lightning discharges mentioned around L104–L112 correspond to Figure 2. Could the authors please clarify this association more clearly?

*This Figure was changed significantly. An image showing the passing of the thunderstorm activity over Germany was added and in the second image, the second flash of lightning was omitted for better readability. This should ensure that the bottom image is understood as showing radio-frequency pulses connected to the strong gamma-ray glow ending. A mention that the first radio-frequency pulses of the lightning flash ending gamma-ray glow occurred above the Milešovka hill was added to the caption.*

[3] Figure 1: Could the authors please explain the meaning of the light and dark gray shaded areas in the caption?

*The following sentence was added to the text about gamma-ray glows for clarity: Temporal positions of those gamma-ray glows are indicated on Figure~\ref{fig:temperature, fig:CP,fig:disdrometer} by different levels of gray: strong gamma-ray glow is indicated by darker grey, whereas weak gamma-ray glow by lighter gray.*

[4] Figure 2: The authors mention that the color scale represents the time since the first lightning, but the relationship between the color change and the passage of time is unclear.
Could you please clarify this relationship?

*The description was not clear, thank you for pointing that up. We had used the term discharge for radio-frequency pulses. A sentence was changed to:Color scale shows time in milliseconds from the first pulse of the flash.*

[5] Figure 3: Would it be possible to indicate the ranges for low-intensity gamma-ray glows and high-intensity gamma-ray glows directly on this figure?

*Figure 3 was removed and the main figure showing both strong and weak gamma-ray glows was moved up (previously Figure 7). Thus, the strong and weak gamma-ray glow should be understandable earlier in the text.*

[6] Figure 4: Could you please describe in the text the difference between high-intensity gamma-ray glows and low-intensity gamma-ray glows?Also, is it possible to deduce the structure (length and/or width) of the electric field in the direction in which electrons are accelerated toward the ground?

If such a deduction is possible, including this information would be beneficial for the reader.

*Sadly, such a deduction is not possible. We cannot know the width of electric field, its orientation, strength, nor its distance from the detectors. We would need more detectors spreadout in a matrix.*

*From the radar data we can guess that the weak gamma-ray glow originated from the cell that passed south of Milešovka. Blitzortung network showed that radio-frequency pulses were detected/spread in a North-East direction from its initiation above the observatory. So, the south cell was not discharged and could be the source of the weak gamma-ray glow. A whole section of weak gamma-ray glow was added.*

[7] Although the percentage of the observed increase is stated, the actual statistical significance is unclear because the error is not shown in Figure 3. Could you please include the error in the counts in Figure 3?

*Figure 3 was removed and the main figure showing both strong and weak gamma-ray glows was moved up (previously Figure 7). It contains errors for each data point.*

[8] 3.4.1 Spectrum: Is the spectrum shown here for the strong gamma-ray glow? If so, is it possible to derive a separate spectrum for the weak gamma-ray glow? According to Figure 7, the weak gamma-ray glow has a wider temporal range than the strong gamma-ray glow. However, the spectra shown here seem to be taken from the weak gamma-ray glow region, possibly because the two are considered distinct.

Alternatively, could it be that the spectrum obtained does not change significantly even if the background (BG) is taken from an earlier time? I would appreciate clarification in the paper on how the strong and weak gamma-ray glows are separated and how the BG affects the derived spectrum.

*The spectrum figure was removed as the second reviewer pointed out that with the time gaps, the data from the spectrometer are unreliable. He also comments on the smoothness of the spectrum. We decided to remove the spectrum from the paper.*

Additionally, Figure 4 refers to low-intensity and high-intensity gamma-ray glows, which I believe correspond to the weak and strong gamma-ray glows, respectively. Is this understanding correct? If so, it would be better to use consistent terminology throughout the paper to avoid confusion.

*Thank you, for pointing that up, a consistency of terminology is now used.*

[9] L126-L128: "LDR and RHO values (Figure 4) show that the height of the thundercloud reflecting back the signal was even smaller, reaching a height of only 2500 m and only 1500 m."

I understand the meaning of the last 2500 m from Figure 4, but I did not understand the meaning of the 1500 m.

Could you please add a brief explanation to your paper?

*It was meant as the measured maximum and at the time of the discharge. The sentence was adjusted:*

*LDR and RHO values (Figure 4) show that the height of the thundercloud reflecting back the signal was even smaller, reaching a height of only 2500 m at the maximum and only 1500 m at the time of the discharge.*

[10] L130–L136: Around these lines, the terms low-intensity gamma-ray glow and high-intensity gamma-ray glow are introduced, but they have not been defined earlier in the text. Could the authors please explain how these terms are derived and provide clear definitions?

*This was a mistake and was meant strong and weak gamma-ray glow. The mention and discussion over strong and weak gamma-ray glows were moved up in the paper, just below the detector description.*

Additionally, please clarify whether these glows correspond to the weaker gamma-ray glow and strong gamma-ray glow mentioned later in the text. The lack of such descriptions makes the connection with Figure 5 unclear.

*The change of order of the text and also the unifying of the terminology should clarify it for the reader.*

[11]L148-149: "This could confirm the previous existence of an area of intensified electric field that aligned the ice crystals and also caused the gamma-ray glow observed on the ground." I am not well sure why this result indicates such a situation. Could you add a brief explanation in the text?

*Thank you for pointing that it is not clear from the text. Following paragraph was added to the text explaining relation between LDR and RHOHV to crystal alignment:*

*In the upper parts of the cloud, the ice crystals are usually columnar in shape. If these crystals are oriented horizontally along the long axis, then Zh is larger and Zv smaller*

*than when the crystals are oriented vertically. Hence, the LDR is higher in the latter case than in the former case. Usually ice crystals are randomly oriented in the scan volume, but when these crystals are exposed to a strong electric field, they arrange along the longer axis vertically making the LDR to increase, and since they are symmetric in the scan direction, it makes the RHOHV to increase as well.*

[12]L162-163: "The data acquisition stopped 3 seconds before the discharge". Why did the acquisition stop? Is this due to pile-up?

*This was due to a software mistake in the spectrometer that messed with its timing. At the time of this event around every half a minute. Although, there are two datasets inside the spectrometer, we still could not recover reliable data for the time of discharge and therefore we left those timestamps empty. Overall the second reviewer deemed the spectrometer data unreliable and we focused only on SEVAN data.*

Technical corrections

[1] Figure 6: This figure does not appear to be referenced in the text.

If it is not essential, perhaps it should be removed. Alternatively, please revise the text to include a reference to it.

*A reference was added to the text, connecting radar data with disdrometer measurements.*

RC2

This manuscript reports the first detection of a gamma-ray glow in continental winter. Winter thunderstorms develop in specific regions, such as the Mediterranean Sea, the Great Lakes, and the northern coast of Japan, and those in the other areas are quite rare. Gamma-ray glows are second-to-minute lasting radiation bursts associated with thunderstorms. Gamma-ray glows associated with winter thunderstorms have been reported from groups in Japan. As far as this reviewer knows, this is actually the first report of a winter gamma-ray glow in continental Europe. Therefore, this manuscript is potentially worth publication. However, a major revision is recommended before the decision.

The biggest issue is the electric field measurement. The authors claim that the gamma-ray glow occurred in a weak electric-field situation. However, the measurement is unreliable as the field mill did not detect the glow-terminating lightning discharge. The Boltek EFM-100 is sensitive to lightning discharges, and the detections of glow-associated discharges have been reported by Chilingarian et al., JGR Atmospheres, 2017, and Chilingarian et al., Atmospheric Research, 2020. The authors described that the Milesovka observatory reported a very close discharge associated with the glow. This reviewer cannot believe that the field mill did not detect the close lightning discharge. The authors should verify the time tagging of the data. If no anomaly is found, the result is unreliable unless the authors present a convincing reason. Then, the descriptions and discussions concerning the field-mill observation should be deleted because the conclusion somewhat contradicts the previous reports, and the authors must avoid a sensitive conclusion with an unreliable measurement.

The second issue is the structure of the manuscript. Because the main focus of this manuscript is gamma-ray glow, the authors should describe the feature of the gamma-ray glow first and then describe the meteorological condition and other measurements, such as the cloud profiler and the field mill. However, the Results section started with a description of the lightning discharge and the explanation for the glow itself was made later in the section. Especially the definition of strong and weak glows should be put earlier.

Detailed comments are listed below.

*Thank you for your thorough reading and deeply impactful suggestions.*

Abstract: please reconsider the statements on the field-mill observation.

*The abstract was rewritten considering not using data from the electric field mill and RT-56 spectrometer based on your suggestions:*

*This study presents the first-ever published detection of two parallel winter continental gamma-ray glows in Central Europe, observed during a rare winter thunderstorm on Milešovka hill, Czechia. The combination of the hill's altitude of 837 m asl and the low altitude of the winter thunderstorm cloud resulted in an observation very near the acceleration region inside the thundercloud. The event was captured using a combination of advanced instruments, including a Ka-band cloud profiler, which enabled detailed analysis of the storm's microphysics, SEVAN large area scintillation detector for monitoring ionizing radiation, and a manned professional meteorological observatory. The radar data indicated the alignment of ice crystals within the cloud, strongly suggesting the presence of a substantial electric field. The findings offer valuable insights into winter thunderstorm dynamics in continental climates, with broader implications for studying high-energy atmospheric physics.*

L.13: Kelley et al. and Kochkin et al. are not the first reports of gamma-ray glows by airborne experiments.

*The sentence was rewritten for clarity: Observations of radiation coming from thunderclouds were made onboard airplanes and balloons (McCarthy and Parks, 1985; Kelley et al., 2015; Kochkin et al., 2017) as well as on the ground at locations*

L.14: The references should be placed right after the place name. Also, it would be better to include the country name along with the place name.

*The sentence was rewritten according to the suggestion: Lomnický štít, Slovakia (Kudela et al., 2017; Chum et al., 2020; Šlegl et al., 2022; Chilingarian et al., 2021c), Musala, Bulgaria (Chilingarian et al., 2021c), Aragats, Armenia (Chilingarian et al., 2021c, b, 2016; Chilingarian and Mkrtchyan, 2012; Chilingarian et al., 2010; Williams et al., 2022), Yangbajing, Tibet, China (Tsuchiya et al., 2012), and Japan (Tsuchiya et al., 2007, 2009, 2011; Torii et al., 2009, 2011; Wada et al., 2021b, a, 2019, 2018, 2023).*

L.23: Does "peak" mean count rate or surface electric field?

*The peak was intended in count rates. the sentence was rewritten: The peak of count rates is detected at the moment of the shortest distance between the detector and the center of the active (charged) region.*

L.26: Please remember that lightning discharges may also terminate MOS-origin glows.

*A mention of MOS process was added: This discharge reduces at least one of the charged regions and makes the RREA or MOS process stop.*

L.42: Please add the unit to "0".

*Unit of MeV was added.*

L.44: This paragraph is somewhat messy and should be rewritten. For example, time-over-threshold should be explained more. Also, statements of pileups and GPS time tagging should be separated.

*The paragraph was divided and expanded:*

*The secondary spectrum of the spectrometer stores information on deposited energies higher than the primary spectrum. This spectrum stores information about the energy of particles in a Time-over-Threshold mode. In this case, it stored the number of time intervals of 25 ns when the peak of the particle's light curve exceeded the threshold matching 4.5 MeV. This spectrum was later calibrated by fitting the peak shape to detected light curve's parts below the threshold and thus obtaining the peak's maximum corresponding to the time over the threshold.*

*The RT-56 spectrometer also calculates the number of pileups per second and contains a GPS module for precise timing. The spectrometer is able to provide a timestamp of up to 4000 particles every second in a selected energy interval. In the case of the winter thunderstorm on February 4, 2022, studied in this paper, the timestamping was set for energies only up to 4.5 MeV.*

L.48: This description is unclear. What is set for 4.5 MeV? According to the description, only below 4.5 MeV seems to be registered, which is not the case.

*The sentence was rewritten: In the case of the winter thunderstorm on February 4, 2022, studied in this paper, the timestamping was set for energies only up to 4.5 MeV.*

L.55: Please specify how to record the photons by SEVAN, event-by-event recording, or only count-rate recording.

*Only count-rate recording. The sentence was added:  The SEVAN detector provides only count rates per second.*

L.61: Please explain how to calibrate the surface field measurement. Typically, it should correspond to 0.1 kV/m in fairweather.

*Electric field data were removed from the paper.*

L.64: Radar parameters should be explained more because the community of high-energy atmospheric physics is not usually familiar with radar meteorology. Also, the radar seems to work with alternate transmission and simultaneous reception mode as it can obtain LDR. It should be explained (not trivial for outside the radar community).

*The radar description was expanded to the following text:*

*Cloud profiler MIRA 35c at the Milešovka station is a vertically oriented Doppler polarimetric radar manufactured by the METEK GmbH company. It operates within Ka-band with a center frequency of 35.12±0.1 GHz and a peak power of 2.5 kW. The temporal resolution is 2 s and vertical resolution includes 509 gates separated by 28.8 m. More technical details are given*

*in Sokol et al. (2020) and Kolmašová et al. (2022). Data provided by the cloud profiler, which are used in this study, are radar reflectivity (Ze), Doppler vertical velocity (V), its spectrum width (W), Linear Depolarization Ratio (LDR), and co-polar correlation coefficient (RHO).*

*Cloud radars differ from conventional operational radars in that they use significantly higher frequencies than operational weather radars (usually working in C-, S- or X-band). This is reflected in the measurement characteristics. Contrary to operational weather radars which record and recognize bigger (precipitation) particles, cloud radars can measure fine particles, such as cloud droplets or ice crystals thereby enabling individual hydrometeor distinction. Further, cloud radars generally have much higher spatial resolution as compared to operational weather radars. In contrast, the measurements of cloud radars are more affected by attenuation by heavy rain and the radar beam reflections follow Mie scattering rather than Rayleigh scattering, which influences the processing of e.g., Ze. The Ka-band cloud radar at the Milešovka station emits energy in the horizontally polarized plane but receives reflections in both the horizontal (Zh; co-channel) and vertical (Zv; cross-channel) planes. The LDR, which is the ratio of Zv to Zh, allows determining the symmetry of the measured object (its shape), which contributes significantly to distinguish the type of a hydrometeor present in a cloud.*

*V determines the velocity of the object in the radial direction and W characterizes the variability of the velocity of moving hydrometeors. Since the measurements are made in the vertical direction and individual hydrometeors have different terminal velocities, high values of W indicate the existence of different hydrometeors in the scan volume. The value of RHO depends on the symmetry/shape of the hydrometeors and high RHO values correspond to symmetric objects.*

*Postprocessing of the data was carried out following the methods described in Sokol et al. (2018). It consists of calculating vertical air velocity (Va), which is then used in the classification of hydrometeors developed and described in the paper.*

L.88: Figure 1 is not enough to determine the thunderstorm system. In fact, frontal systems are associated with a drop in temperature, but this is also the case for downbursts. The easiest way is to refer to radar observation and weather charts. Is the operational radar data available? If the authors would like to put a stress on meteorological conditions, these data are necessary.

*Additional information on the existence of the cold front was added showing CAPPI 2km from the C-band operational radar of the Czech Hydrometeorological Institute, as required. We cannot talk about the existence of a downburst in this case because there were no extreme winds near the ground that would cause damage.*

*The following paragraph was added with two figures:*

*The passage of the cold front is confirmed by the synoptic map (Fig. 1) and the radar reflectivity at the CAPPI 2km level (Fig. 2) from the operational radar of the Czech Hydrometeorological Institute. The front was moving from the northwest and the radar reflectivity (Fig. 2) shows that there were several storm centers on the front accompanied by lightning.*

L.90: A plot with a wider area than Figure 2 is needed to understand this paragraph. Also, flash rate is important (please be careful with the definition of flashes.)

*Another image of the whole thunderstorm was added to Figure 5 (now). Also, a flash rate of 7.5 flashes/hour was calculated and added to the text.*

Figure 2: This figure is somewhat meaningless. The colors should be the same for the two data sets. Also, a scale is needed to emphasize the distance between discharge points and the observatory. Also, please confirm the definition of flash. EUCLID and BLITZ only provide the source position of radio-frequency pulses, which often correspond to lightning currents (or strokes). A lightning flash is a collection of lightning currents.

*Thank you for pointing up the confusing terms. An image of a passing thunderstorm to the Milešovka hill was added. The image of the flash at Milešovka was changed for readability and thde second flash 3 minutes later was removed.*

*The caption of the Figure was rewritten:*

*"Top: Radio-frequency pulses detected by Blitzortung (black-bordered circles) as the thunderstorm passed Germany and entered Czechia. Color scale shows time in hours from the first flash at 17:51:01. Milešovka event happened at the final stage of the thunderstorm. Bottom: Radio-frequency pulses detected by Blitzortung (black-bordered circles) and EUCLID (red-bordered circles) at the time of the end of the strong gamma-ray glow (23:20:08.147-187). Color scale shows time in milliseconds from the first pulse of the flash. Both networks indicate the beginning of the gamma-ray ending flash (its first radio-frequency pulse) very near the Milešovka station depicted by a yellow-bordered circle."*

*The meaning of this Figure is to show that the first flash started right at the top of Milešovka, where the detectors are positioned.*

L.93: The first thing to do here is to show the gamma-ray glow. Especially the presentation and definition of strong and weak glow should be put first. Otherwise, the discussion is quite missing and unclear.

*The whole section about gamma-ray glows was placed after the detectors' description.*

L.98-99: This explanation is not convincing but connected to the main conclusion of this manuscript. Please consider removing or carefully verifying this.

*The data from the electric field mill were removed from the paper.*

Figure 3: There are several missing data for RT-56. This leads us to interpret it as unreliable. In fact, the discussion can be made only with the SEVAN data. Please reconsider whether RT-56 data is included or not. My feeling is that RT-56 should work without GPS signals. Also, please specify that the SEVAN data is non-coincidence or coincidence for the top layer.

*Data from the RT-56 were removed according to your suggestion.*

L.110: Please confirm the definition of flash.

*The whole paragraph was rewritten to use the terms radio-frequency pulses and flash as a group of pulses.*

L.113: Please show the summary of Popova et al. if the analysis is essential for this manuscript.

*This sentence was removed and significant data was adopted to this paper.*

L.114-121: Please entirely reconsider this paragraph.

*The paragraph was removed, again due to unreliable data from the electric field mill.*

L.125: Ka-band is very sensitive to attenuation in strong precipitation. In such cases, the measurement of cloud top height is quite challenging. Also, please show the actual value of cloud top height estimated by the geostationary satellite.

*In this particular case, the attenuation is probably not very significant, as the cloud top height determined by the radar is approximately the same as that determined by the Meteosat Second Generation satellite measurements estimated to be 3 km above the surface by the infrared channel 10.8 µm at 23:20 UTC and temperature profile measured by aerological sounding in Prague at 00 UTC at the Praha-Libuš station situated about 80 km southwards from the Milešovka station.*

*Following sentences were added to the text:*

*The cloud top height was also estimated using Meteosat Second Generation satellite (https://www.eumetsat.int/meteosat-second-generation). Using the brightness temperature of the infrared channel 10.8 µm at 23:20 UTC and the temperature sounding measurements at the Praha-Libuš station (80 km southwards from the Milešovka observatory) at 00:00 UTC on February 5, 2022, the cloud top height was approximately 3 km above the surface (Popová et al., 2023).*

L.129: The definition of the low-intensity glow should be placed earlier.

*The whole section about gamma-ray glows was placed after the detectors' description.*

L.135: Hail can be observed by the disdrometer. The result of the disdrometer should be referred to here.

*Indeed. Reference was added.*

L.137-149: This description should be placed earlier or the first of Section 3.

*The whole section about gamma-ray glows was placed after the detectors' description.*

L.141: Please explain how to sense the vertically aligned ice crystals with LDR and RHOHV.

*Following text was added:*

*In the upper parts of the cloud, the ice crystals are usually columnar in shape. If these crystals are oriented horizontally along the long axis, then Zh is larger and Zv smaller than when the crystals are oriented vertically. Hence, the LDR is higher in the latter case than in the former case. Usually ice crystals are randomly oriented in the scan volume, but when these crystals are exposed to a strong electric field, they arrange along the longer axis vertically making the LDR to increase, and since they are symmetric in the scan direction, it makes the RHOHV to increase as well.*

L.157: Please define "high-energy" photons.

*It was meant as photons with E higher than 3 MeV, but this sentence is not necessary as is removed.*

L.161-163: This description should be explained more. Again, the discussion with RT-56 data seems to be unreliable.

*Data from the RT-56 were removed according to your suggestion.*

L.174: What are the wind speed and cloud base information used for? Please be careful with the cloud base information because the ceilometer cannot measure it during heavy precipitation.

*The sentence was rewritten:*

*If the strong gamma-ray glow had not been interrupted by the discharge and had continued following the Gaussian curve, the duration of the peak would have been roughly 2.5 minutes, which aligns with the previous observations as the wind speed of 10 m/s does not deviate from regular wind speed during thunderstorms (with higher speed the acceleration region moves faster and thus creates a narrower peak). Also, the peak count rate of 5552 counts/s for SEVAN top channel with subtracted background is reasonable as the cloud base height was less than 30 meters (the lower the accelerating region the lower the attenuation of the radiation). There were even higher detected count rates of SEVAN top channel on Lomnický štít, Slovakia (\citep{chum2020})*

*The precipitation was not heavy and this type of ceilometer also measures during rainfall.*

Figure 6: The intent of this diagram is unclear. Also, it would be better to unify the range of the vertical axis. Please explain solid precipitation and no hydrometer.

*This figure was removed and partially included in the figure showing meteorological data. The data were summed to groups in order to support radar hydrometeor classification (graupel+hail, ice+snow, cloud+raindrops). By applying the same range for the graphs the readability was lowered. We believe it should show the time evolution relative to the gamma-ray glows.*

Figure 7: A more precise explanation of Gaussian fittings should be made in the main text. Also, how do we define the center time if a double Gaussian function is used for the fitting? Again, the discussion can be done without RT-56.

*A paragraph about Gaussian fitting was added to the beginning of the results and discussion about gamma-ray glows:*

*As the function F (t) to fit temporal evolution of count rate a Gaussian curve was selected as in (Wada et al., 2021b):105*

*F (t) =SUM2Xi=1ai exp[−(t − µi)^2/2σi^2]+ c , (1)*

*where t is time, a is the peak count rate, µ is the peak time, σ is the standard deviation, and c is the background count rate.*

*Gaussian function was selected only because the time series of count rates roughly follow this line. However, here is no physical*

*explanation. This function has a benefit of easy calculation of Full duration in half maximum (F DHM ≈ 2.355 σ)*

*Data from the RT-56 were removed according to your suggestion.*

Figure 8: The high-energy part (>3 MeV) is not smooth, different from the previous report (Wada et al., PRR, 2021). Please explain why such a structure can be seen. Also, I have

doubts about the accuracy of the energy above 4.5 MeV (determined by time-over-threshold). How to calibrate the high-energy domain?

*Data from the RT-56 were removed according to your suggestion.*

L.187: My feeling is that this manuscript is just a case report rather than a scientific paper. Please take the comments above into account and include more discussions.

*We enhanced the discussion of the weak gamma-ray glow.*

**Citation**: https://doi.org/10.5194/egusphere-2024-3075-RC2